# Neurological Prognostic Factors in Hospitalized Patients with COVID-19

**DOI:** 10.3390/brainsci12020193

**Published:** 2022-01-30

**Authors:** Leszek Drabik, Justyna Derbisz, Zaneta Chatys-Bogacka, Iwona Mazurkiewicz, Katarzyna Sawczynska, Tomasz Kesek, Jacek Czepiel, Pawel Wrona, Joanna Szaleniec, Malgorzata Wojcik-Bugajska, Aleksander Garlicki, Maciej Malecki, Ralph Jozefowicz, Agnieszka Slowik, Marcin Wnuk

**Affiliations:** 1Department of Pharmacology, Jagiellonian University Medical College, 16 Grzegorzecka St., 31-531 Krakow, Poland; leszek.drabik@uj.edu.pl; 2John Paul II Hospital, 80 Pradnicka St., 31-202 Krakow, Poland; 3Department of Neurology, University Hospital in Krakow, 2 Jakubowskiego St., 30-688 Krakow, Poland; justyna.derbisz@gmail.com (J.D.); zaneta_chatys@wp.pl (Z.C.-B.); imazurkiewicz@su.krakow.pl (I.M.); katarzyna.sawczynska@gmail.com (K.S.); tomaszkesek11@gmail.com (T.K.); wronapawelmichal@gmail.com (P.W.); agnieszka.slowik@uj.edu.pl (A.S.); 4Department of Neurology, Jagiellonian University Medical College, 2 Jakubowskiego St., 30-688 Krakow, Poland; 5Department of Infectious Diseases, University Hospital in Krakow, 2 Jakubowskiego St., 30-688 Krakow, Poland; czepiel.jacek@gmail.com (J.C.); agarlicki@gmail.com (A.G.); 6Department of Infectious and Tropical Diseases, Jagiellonian University Medical College, 2 Jakubowskiego St., 30-688 Krakow, Poland; 7Department of Otorhinolaryngology, University Hospital in Krakow, 2 Jakubowskiego St., 30-688 Krakow, Poland; joanna.szaleniec@uj.edu.pl; 8Department of Otorhinolaryngology, Jagiellonian University Medical College, 2 Jakubowskiego St., 30-688 Krakow, Poland; 9Department of Internal Medicine and Gerontology, Jagiellonian University Medical College, 2 Jakubowskiego St., 30-688 Krakow, Poland; mwojcik@su.krakow.pl; 10Department of Internal Medicine and Gerontology, University Hospital in Krakow, 2 Jakubowskiego St., 30-688 Krakow, Poland; 11Department of Metabolic Diseases and Diabetology, Jagiellonian University Medical College, 2 Jakubowskiego St., 30-688 Krakow, Poland; maciej.malecki@uj.edu.pl; 12Department of Metabolic Diseases and Diabetology, University Hospital in Krakow, 2 Jakubowskiego St., 30-688 Krakow, Poland; 13Department of Neurology, University of Rochester Medical Center, Rochester, 601 Elmwood Ave, Rochester, NY 14642, USA; Ralph_Jozefowicz@urmc.rochester.edu

**Keywords:** COVID-19, SARS-CoV-2 infection, neurological symptoms, stroke, prognosis, in-hospital mortality, oxygen therapy

## Abstract

We aimed to search whether neurological symptoms or signs (NSS) and the MEWS (Modified Early Warning Score) score were associated with in-hospital mortality or oxygen requirement during the first 14 days of hospitalization in COVID-19 patients recruited at the University Hospital in Krakow, Poland. The detailed clinical questionnaires on twenty NSS were either filled out by patients prospectively or retrospectively assessed by neurologists based on daily medical records. NSS were considered high or low-risk if they were associated with increased or decreased mortality in the univariable analysis. This cohort study included 349 patients with COVID-19 (median age 64, interquartile range (51–77), women 54.72%). The presence of high-risk NSS (decreased level of consciousness, delirium, seizures, and symptoms of stroke or transient ischemic attack) or its combination with the absence of low-risk NSS (headache, dizziness, decreased mood, and fatigue) increased the risk of in-hospital mortality in SARS-CoV-2 infection 3.13 and 7.67-fold, respectively. The presence of low-risk NSS decreased the risk of in-hospital mortality in COVID-19 patients more than 6-fold. Death in patients with SARS-CoV-2 infection, apart from NSS, was predicted by older age, neoplasm, and higher MEWS scores on admission. High-risk NSS or their combination with the absence of low-risk NSS increased the risk of oxygen requirement during hospitalization in COVID-19 patients 4.48 and 1.86-fold, respectively. Independent predictors of oxygen therapy during hospitalization in patients with SARS-CoV-2 infection were also older age, male sex, neoplasm, and higher MEWS score on admission.

## 1. Introduction

The Severe Acute Respiratory Syndrome coronavirus 2 (SARS-CoV-2) infection caused a significant burden for public health systems, with over 5.5 million deaths worldwide as of 22 January 2022 [1]. Mortality among patients admitted to hospital due to this disease was found to be nearly three times higher when compared to hospitalized cases with influenza [2]. Additionally, more than half of young patients with Coronavirus Disease 2019 (COVID-19) required oxygen therapy during hospitalization [3]. The COVID-19 pandemic also resulted in the increased incidence of mental health disorders, such as anxiety, depression, and insomnia [4]. The broad variety of neurological symptoms in the course of SARS-CoV-2 infection ranged from mild manifestations, including myalgia, headache [5], fatigue, dizziness, and anosmia, to more severe presentations, such as seizures, stroke, and encephalopathy [6]. Involvement of both the central and peripheral nervous systems was common within the first 14 days of the SARS-CoV-2 infection [7]. Moreover, a substantial number of patients experienced persistent neurological symptoms within 90 days after discharge from the COVID-19 ward [8].

Therefore, the need for simple and reliable prognostic factors that could be helpful in predicting the course of SARS-CoV-2 infection during hospitalization still exist [9]. Previous studies pointed to many clinical [10] and paraclinical [11] findings as possible risk factors for deterioration of patient or death. For example, in an Italian study of 87 patients with COVID-19, a high Monocyte Distribution Width (MDW) value emerged as the prognostic factor of fatal outcome with sensitivity 0.75 and specificity 0.70 [12] that reflected hyperinflammation mediated by monocyte/macrophage subsets [13,14,15,16]. In another study, the presence of neurological symptoms that were associated with a poorer prognosis was predicted by low lymphocyte count and increased lactate dehydrogenase and interleukin-6 [17]. However, a recent German study showed that self-reported patients’ symptoms after COVID-19 corresponded well with the objective findings in the neuropsychological testing, including fatigue, sleep disturbances, anxiety, and depression, especially among hospitalized patients [18]. Moreover, neurological manifestations became increasingly noted in the course of COVID-19, even in critically ill patients [19], and some of them, such as cerebrovascular disease, were associated with worse prognoses [20]. Our previous study revealed that different neurological symptoms increased in-hospital mortality, whereas others exhibited a protective role in this issue [21]. However, the role of the combination of different neurological symptoms in mediating the risk of death or requirement for oxygen therapy was not sufficiently investigated.

Many prognostic scales corresponded with mortality risk among patients with COVID-19, such as Pneumonia Severity Index [22] or CURB-65 [23], among others. In Poland, where more than 4 million cases with SARS-CoV-2 infection were noted as of 22 January 2022 [1], the easy to obtain Modified Early Warning Score (MEWS) was extensively used in clinical practice in accordance with national COVID-19 guidelines presented at the beginning of the pandemic [24]. However, there is limited data on the prognostic significance of the MEWS score in patients with COVID-19, and available studies produced divergent results [25,26,27].

Therefore, the aim of the current study was to search whether the combination of easy to identify neurological symptoms, prior central nervous system (CNS) diseases, including stroke and MEWS score, was associated with in-hospital mortality or the requirement of oxygen therapy among hospitalized patients with COVID-19.

## 2. Materials and Methods

### 2.1. Patients

This cohort study included 349 patients with COVID-19 hospitalized between March 2020 and February 2021 at five departments of the University Hospital in Krakow, Poland: Neurology (42.69%, n = 149), Metabolic Diseases and Diabetology (31.23%, n = 109), Infectious Diseases (16.62%, n = 58), Internal Medicine (5.73%, n = 20), and Otorhinolaryngology (3.72%, n = 13). The SARS-CoV-2 infection was diagnosed based on the positive result of real-time reverse transcription-polymerase chain reaction (97.99%, n = 342) or antigen test (2.01%, n = 7) from a nasopharyngeal swab. The following inclusion criteria were used in the study, being the same as causes of hospitalization: low blood saturation (≤92%), dyspnea, chronic disease that needed hospital treatment, or no possibility for social isolation. We excluded patients below the age of 18 and those who required mechanical ventilation and intubation during hospital admission (cardiopulmonary arrest, lost airway or jeopardized airway, respiratory distress with respiratory rate >30/min, or hypoxemia SpO_2_ < 93% on room air and PaO_2_:FiO_2_ < 300 mmHg that progressively became worse besides 2-hour high-flow oxygen therapy [28]).

We gathered the data on basic demographics, concomitant chronic diseases with special emphasis on cardiovascular and neurological disorders, COVID-19 symptomatology, and blood parameters previously associated with poorer COVID-19 prognosis, which were troponin I [29] and D-dimer levels [30]. Central nervous system diseases included the following: the history of stroke, epilepsy, prior traumatic brain injury, tumor, parkinsonian syndrome, and dementia. All data were gathered retrospectively in a specially designed database that was fulfilled by neurologists based on medical records in the hospital system.

### 2.2. Neurological Symptoms and Sings

The presence of neurological symptoms in the course of SARS-CoV-2 infection was evaluated prospectively (59.03%, n = 206) or retrospectively (40.97%, n = 143) as described previously [21]. In brief, the detailed clinical questionnaires on the presence of 20 neurological symptoms or signs (NSS), including 12 symptoms (headache, dizziness, decreased mood, memory or concentration difficulties, fatigue, visual disturbances, anosmia, ageusia, muscle weakness, myalgia, paresthesia, increased sweating) and 8 signs (decreased level of consciousness, delirium, ataxia, seizures, stroke/TIA, autonomic disturbances defined as diarrhea, arterial hypotension <90/60 mmHg or tachycardia >100/min), were either filled out by patients prospectively during the first 14 days of hospitalization or retrospectively assessed by neurologists based on daily medical records in the hospital database. Neurological symptoms or signs, such as headache, dizziness, decreased mood, or fatigue, if not reported by patients, were considered absent. Neurological symptoms and signs that were associated with increased mortality in the univariable analysis were considered high-risk NSS, whereas those that decreased the risk of death in the same analysis were defined as low-risk NSS. From such differentiation of NSS excluded were arterial hypotension and tachycardia as these parameters were dependent on the hemodynamic state of the patient that in turn was reflected by MEWS score on admission. The main study endpoints were the requirement of oxygen therapy or death during hospitalization.

Each patient followed prospectively gave informed medical consent that was either written or verbal in the presence of two witnesses. The study was approved by the Bioethics Committee of the District Medical Council in Krakow (opinion number 143/KBL/OIL/2020) and conducted in accordance with the Declaration of Helsinki.

### 2.3. Statistics

We assumed a 25% difference in the prevalence of NSS based on Brieghel et al. [31]. For such a difference or greater, a minimum sample size of 27 patients in each group was required to achieve an α level of 0.05 and statistical power of 0.80. Patients were stratified according to outcome (death, oxygen therapy) and a group of NSS. We presented categorical values as counts and percentages, and continuous values as means ± standard deviations (SD) or medians (interquartile ranges, IQR), as appropriate. We assessed the distribution normality of continuous variables with the Shapiro–Wilk test and compared, by Student’s *t*-test or by Mann–Whitney U test, as appropriate. Categorical variables were assessed with a chi-square test. To identify predictors of death and oxygen therapy, we used multivariable logistic regression with backward stepwise elimination. All variables that showed association with outcome in the univariable model with *p* < 0.05 and showed no substantial correlation with other independent variables (r > 0.5) were included in a model. Non-linear variables were log-transformed. Models for death included age, prior CNS disease, diabetes mellitus, chronic kidney disease stage ≥3, neoplasm, MEWS score, troponin I, D-dimer, and (1) high-risk NSS (model A), (2) low-risk NSS (model B), and (3) high-risk/or absence of low-risk NSS (model C). The multivariable models (1–3) for oxygen therapy also included hypertension and excluded chronic kidney disease stage ≥3. To assess the goodness of fit, Akaike information criterion and the Wald test were assessed. For each model, the area under the receiver operating characteristic curve (AUC) with 95% confidence intervals (CI) was used to assess predictive accuracy. For each group of NSS, AUC with 95% CI, accuracy, sensitivity, specificity, positive and negative predictive value together with positive likelihood ratio was assessed. Probability values ≤0.05 were considered statistically significant. The data were evaluated using STATISTICA (version 13.0, Statsoft Inc., Tulsa, OK, USA).

## 3. Results

The data that confirm the results of the current study are available from the corresponding author upon reasonable request.

### 3.1. Patient Characteristics

The cohort of 349 patients (median age 64, interquartile range (51–77), women 54.72%) due to COVID-19 is presented in Table 1. Patients were admitted from home (60.74%, n = 212), other hospitals (31.52%, n = 110), nursing homes (6.88%, n = 24), and institutional isolation (0.86%, n = 3). The median time from the first positive nasopharyngeal swab test for SARS-CoV-2 to hospitalization was 1 (1–3) day, and the median time from the onset of COVID-19 symptoms to hospital admission was 5 (3–8) days.

### 3.2. Mortality

Patients who died during hospitalization (9.46%, n = 33) were older, had a higher prevalence of diabetes mellitus, prior CNS disease, neoplasm, chronic kidney disease stage 3, and higher MEWS score, troponin I, and D-dimer compared to survivors (Table 1). Based on the univariable analysis concerning the risk of death, high-risk NSS comprised the following: a decreased level of consciousness, delirium, seizures, and symptoms of stroke or transient ischemic attack (TIA). Conversely, low-risk NSS included headache, dizziness, decreased mood, and fatigue that were reported by patients (Appendix A). High-risk NSS were nearly 4-fold more frequent, and low-risk NSS were more than 2-fold less frequent in this group than in COVID-19 survivors (Table 1, Figure 1A). These results were similar when we separately analyzed patients assessed prospectively and retrospectively; however, there were more deaths in the group collected retrospectively (see Appendix A). There was also accordance with the findings of the study coming from the whole group when patients from different departments were analyzed separately according to the presence of high-risk NSS (see Appendix A).

In the multivariable regression model, death was predicted by age, neoplasm, MEWS score, high-risk NSS, and prior CNS disease (model A), or low-risk NSS (model B), or high-risk/or absence of low-risk NSS (model C) (Table 2). Model C was the best-fitted model based on AIC values. The AUC of high-risk NSS and low-risk NSS for predicting death was 0.756 (95% CI 0.659–0.853) and 0.697 (95% CI 0.572–0.822), respectively (Appendix A).

### 3.3. Oxygen Therapy

Patients requiring oxygen therapy during hospitalization (n = 218, 62.46%) were older and more often males, had a higher prevalence of hypertension, diabetes mellitus, neoplasm, prior CNS disease, and had higher MEWS score, troponin I, and D-dimer compared with the remainder. On admission, they were characterized with a higher frequency of fever, cough, and dyspnea, and lower frequency of sore throat. On day 14 since the onset of hospitalization, 62 (17.12%) patients required oxygen therapy, including 13 (3.72%) mechanically ventilated. We observed a more than five-fold increase in high-risk NSS during hospitalization in the group requiring oxygen therapy (Table 1, Figure 1B).

The independent predictors of oxygen therapy during hospitalization were age, male sex, neoplasm, MEWS score, and high-risk NSS (model A) or high-risk / or absence of low-risk NSS (model C) (Table 3). Model A yielded the lowest AIC values. The AUCs of high-risk NSS for predicting oxygen therapy was 0.630 (95% CI: 0.572–0.689) (Appendix A).

### 3.4. The Severity of Neurological Symptoms and Signs

#### 3.4.1. High-Risk NSS

Almost one in four (n = 77, 22.06%) patients had one or more high-risk NSS, namely decreased level of consciousness (n = 57, 16.33%), delirium (n = 24, 6.88%), seizures (n = 8, 2.30%), and stroke or TIA symptoms (n = 35, 10.03%) (Appendix A, Figure 2A). There were 36 patients with more than one high-risk NSS, including n = 28 (7.98%) with two, n = 6 (1.72%) with three, and n = 2 (0.57%) with four high-risk NSS, respectively.

Patients with high-risk NSS were older (76 (66–84) vs. 61 (49–71) years, *p* < 0.001), had a higher prevalence of hypertension (73.52 vs. 55.22%, *p* = 0.002), diabetes mellitus (40.26 vs. 22.01%, *p* = 0.001), ischemic heart disease (28.57 vs. 13.43%, *p* = 0.002), prior CNS disease (63.64 vs. 17.72%, *p* < 0.001), tachycardia (40.26 vs. 26.87%, *p* = 0.024), and arterial hypotension (45.45 vs. 11.19%, *p* < 0.001), a higher requirement for oxygen therapy with a non-re-breather mask on admission (46.75 vs. 10.48%, *p* < 0.001); higher MEWS score (MEWS ≥3, 13.33 vs. 2.61%, *p* < 0.001), troponin I (13.98 (5.80–33.47) vs. 5.00 (2.72–11.88) mg/dL, *p* < 0.001), and D-dimer (1.45 (0.72–3.23) vs. 0.63 (0.41–1.23) mg/L, *p* < 0.001) compared to the remainder (Appendix A). The prevalence of sore throat (5.19 vs. 13.93%, *p* = 0.024) and loss of appetite (7.80 vs. 27.79%, *p* < 0.001) as a first COVID-19 symptom was lower in the group with high-risk NSS (Appendix A).

Patients with any high-risk NSS had higher mortality compared with those who did not manifest symptoms (28.57 vs. 3.73%, *p* < 0.001) (Figure 1A). The presence of at least one high-risk NSS was associated with more than a 3-fold increase in the risk of death in the multivariable model (Table 2). With the increasing number of high-risk NSS, the prognosis deteriorated, with 50% or more risk of death when three or four symptoms coexisted in one patient (Figure 2B). Any high-risk NSS were associated with an increased need for oxygen therapy (88.31 vs. 54.31%, *p* < 0.001) and borderline with high-concentration oxygen therapy via a non-rebreather mask (46.75% vs. 10.48%, *p* = 0.08) compared with patients without those symptoms (Appendix A, Figure 1B). Stroke or TIA during COVID-19 analyzed as a single NSS was not associated with the risk of death (OR = 2.59, 95% CI: 0.78–8.65, *p* = 0.122) or oxygen therapy requirement (OR = 2.09, 95% CI: 0.74–5.93, *p* = 0.164) in the multivariable analyses (see Appendix A).

#### 3.4.2. Low-Risk NSS

A total of 246 patients (70.49%) had at least one low-risk NSS: headache (n = 130, 37.24%), dizziness (n = 78, 22.35%), decreased mood (n = 44, 41.26%), and fatigue (n = 200, 57.31%). More than one low-risk NSS was found in 187 (53.58%) patients (Appendix A, Figure 2A).

Patients with low-risk NSS were younger (61 (49–73) vs. 70 (57–79) years, *p* = 0.008), had a lower prevalence of hypertension (55.69 vs. 69.51%, *p* = 0.027), prior CNS disease (13.41 vs. 30.49%, *p* < 0.001), lower troponin I (5.46 (2.79–12.94) vs. 10.18 (4.53–21.94) mg/dL, *p* = 0.001), and D-dimer (0.66 (0.43–1.27) vs. 1.00 (0.43–1.82) mg/L, *p* = 0.042) compared with the remainder. The prevalence of asthma/chronic obstructive pulmonary disease (7.32 vs.1.22%, *p* = 0.041), fever (63.01 vs. 48.78%, *p* = 0.024), cough (67.07 vs. 54.88%, *p* = 0.047), sore throat (15.04 vs. 6.10%, *p* = 0.036), loss of appetite (38.62 vs. 10.98%, *p* < 0.001), dyspnea (54.07 vs. 35.37%, *p* = 0.034), and abdominal pain (23.58 vs. 8.54%, *p* = 0.003), as first COVID-19 symptoms, was higher in the low-risk NSS group (Appendix A).

Patients with any low-risk NSS had lower mortality (3.25 vs. 15.85%, *p* < 0.001) compared with those who did not have those symptoms or were unable to report them (Figure 1A). The presence of at least one low-risk NSS was associated with more than a 6-fold decrease in the risk of death in the multivariable model but not oxygen therapy (Table 2 and Table 3).

## 4. Discussion

To our knowledge, the current study is the first to show that a combination of neurological symptoms, rather than a single symptom, may predict in-hospital mortality in patients with COVID-19. We found that a combination of high-risk NSS and the absence of low-risk NSS, increased the risk of in-hospital death nearly 8-fold in patients with SARS-CoV-2 infection. It was previously noted that isolated neurological symptoms, such as decreased level of consciousness [32], delirium [33], or stroke symptoms [34], were associated with increased mortality in COVID-19 patients. The latest literature review showed that any neurological manifestation of SARS-CoV-2 infection was correlated with a higher mortality rate [35]. A similar conclusion came from a prospective study of Egyptian patients with COVID-19, demonstrating a significant association between neurological events and both a severe course of the disease at onset and higher mortality [7]. On the other hand, in an Italian study of 901 COVID-19 patients, after adjustment for age, sex, and comorbidities, isolated neurological symptoms increased the chance of survival, but nearly one-third of these symptoms comprised dysgeusia, anosmia, or syncope [36]. Of these, anosmia was associated with a nearly 7-fold decrease in the risk of death among more than 11,000 patients with COVID-19 included in a large meta-analysis of 26 studies [37]. A recent Brazilian retrospective cohort study that included 613 patients with COVID-19 showed that, apart from older age and a requirement for mechanical ventilation, patients with encephalopathy had a significantly higher risk of death, but those with anosmia had a lower risk of dying [6]. Many other studies confirmed the protective role of another neurological symptom, namely headache, in terms of the risk of death due to SARS-CoV-2 infection [21,38,39]. Therefore, it appears that different NSS exhibit a diverse correlation with the mortality risk in COVID-19 patients. Moreover, not only does the quality of these symptoms but also their quantity play an important role, with more high-risk NSS, including seizures, delirium, decreased level of consciousness, or stroke or TIA symptoms, leading to an increased risk of death in COVID-19 patients. What is worth emphasizing is that our models for evaluating high and low-risk NSS were also accurate based on AIC/AUC ratios with high negative predictive values.

Our study also revealed that high-risk NSS were independent predictors of oxygen therapy requirement during hospitalization. The literature to date regarding this issue is sparse. A previous study including nearly 900 patients from the NHS Trust hospital in London reported that the main indicators of poor outcomes in COVID-19 were age and oxygenation status in the emergency room [40]. An analysis of the Brazilian Ministry of Health Database encompassing hospitalized cases with SARS-CoV-2 infection showed that blood oxygen saturation levels below 95% increased the risk of death by 1.27-fold [41]. On the other hand, in a Korean study of more than 5000 patients with COVID-19, logistic regression analyses showed that male sex and older age increased the risk of oxygen therapy requirements during hospitalization [42], similar to the findings in our study. A recent analysis from one of the Indian tertiary centers revealed that patients with SARS-CoV-2 infection who died in the course of the disease received high flow nasal cannula oxygen nearly four times more often than survivors [43].

Our study demonstrates the important prognostic role of the MEWS score in patients with COVID-19 since a score of three or more points was associated with a higher risk of death or oxygen demand during hospitalization in each statistical model tested. Patients with respiratory and systemic symptoms of COVID-19 with a MEWS score lower than three were perceived as stable according to previous national guidelines [24]. These findings were similar to our study, in which nearly one-fourth of patients with SARS-CoV-2 also had prior CNS disease. However, previous studies on the MEWS score as a prognostic factor in COVID-19 patients produced contradictory results. A small Chinese study showed that the REMS (Rapid Emergency Medicine Score), which included, in addition, age and oxygenation, correlated better with mortality risk than the MEWS score [44]. A larger Chinese study comparing five different early warning scores in COVID-19 patients showed that the MEWS, when measured on admission, did not predict death risk [25]. On the other hand, a Turkish study revealed that the MEWS score calculated in the emergency room effectively predicted death in the forthcoming 28 days in patients with COVID-19 who required hospitalization [26]. The results of this study were in line with findings from a large cohort of 1000 patients from five Dutch hospitals [45]. A retrospective analysis of data from Wuhan showed that the MEWS was accurate in predicting in-hospital mortality in elderly patients with SARS-CoV-2 infection [46]. A small Polish study also showed that the MEWS predicted a severe course of SARS-CoV-2 infection [47]. The MEWS, which appears to be reliable and easy to assess, was found to be useful in predicting neurological worsening in a cohort of more than 7000 neurocritically ill patients with COVID-19 [48]. However, a recent Indian study showed that not many rating scales, including the MEWS, were found to be useful in identifying patients with COVID-19 who are at risk for clinical worsening [27].

In our study, we were able to show that patients with COVID-19 and prior CNS disease had more than a five-fold increase in the risk of in-hospital mortality. A previous study performed in a group of nearly 600 COVID-19 patients from four hospitals in Ohio found that, although a pre-existent major neurological disease, such as dementia, stroke, or epilepsy, increased the risk of death by a factor of two, it was not an independent predictor of mortality in the multivariate analysis [49]. The same conclusions could be drawn from a small Korean study of COVID-19 patients in an intensive care unit where a history of neurological disease did not affect mortality, in contrast to new-onset neurological complications [50]. On the other hand, the presence of chronic neurological comorbidities, such as a prior stroke with long-term sequelae, cognitive disorders, or neuromuscular and spinal diseases, independently predicted the risk of death in a Spanish cohort of patients with SARS-CoV-2 infection [51]. Most of our cohort with prior CNS disease consisted of patients with pre-existent stroke. A history of stroke was found to independently predicate in-hospital mortality in a large cohort of more than 3000 patients with COVID-19 hospitalized in the Mount Sinai Health System in New York [52]. Similarly, in a Romanian study where nearly 8% of COVID-19 patients had a prior stroke, the presence of a preexisting neurological disorder was highly correlated with SARS-CoV-2 infection severity [53].

Our study has important limitations. First, the number of patients was rather small. The cohort of patients was mixed, and only 59% prospectively filled out clinical questionnaires. Second, information on low-risk NSS was obtained only voluntarily from our patients. Third, selection bias was present in some patient cohorts, particularly in the Departments of Otorhinolaryngology and Internal Medicine, where cohort sizes were small. There were also a small number of patients with mild SARS-CoV-2 infection in the first wave of the pandemic who were hospitalized only due to the lack of the ability to socially isolate. Fourth, all patients in our study were Caucasian; therefore, the results might not be generalizable to other ethnicities, as was shown in a large meta-analysis [54]. Fifth, measuring serum [12] or cerebrospinal fluid (CSF) [55] biomarkers could potentially improve the prognostic ability of our COVID-19 models, but based on the AIC/AUC values, our predictive models appear to be accurate. Sixth, neuroimaging, CSF analyses, and electroencephalography were performed only in the minority of our patients and guided by the clinical history and neurological examination. Previous studies reported that certain neuroimaging features, such as cerebral microbleeds or leukoencephalopathy, were associated with higher mortality and worse outcome on discharge [56]. Therefore, future studies are needed to answer the question of whether additional testing such as neuroimaging could improve our ability to predict the course of COVID-19.

## 5. Conclusions

The presence of high-risk NSS (decreased level of consciousness, delirium, seizures, and symptoms of stroke or transient ischemic attack) or its combination with the absence of low-risk NSS (headache, dizziness, decreased mood, and fatigue) increases the risk of in-hospital mortality in SARS-CoV-2 infection 3- and nearly 8-fold, respectively. The presence of low-risk NSS decreases the risk of in-hospital mortality in COVID-19 patients more than 6-fold. Death in patients with SARS-CoV-2 infection, apart from NSS, is predicted by older age, neoplasm, and higher MEWS scores on admission. High-risk NSS or the combination of high-risk or absence of low-risk NSS increases the risk of oxygen requirement during hospitalization in COVID-19 patients nearly 5- and 2-fold, respectively. Independent predictors of oxygen therapy during hospitalization in patients with SARS-CoV-2 infection are also older age, male sex, neoplasm, and higher MEWS score on admission.

## Figures and Tables

**Figure 1 brainsci-12-00193-f001:**
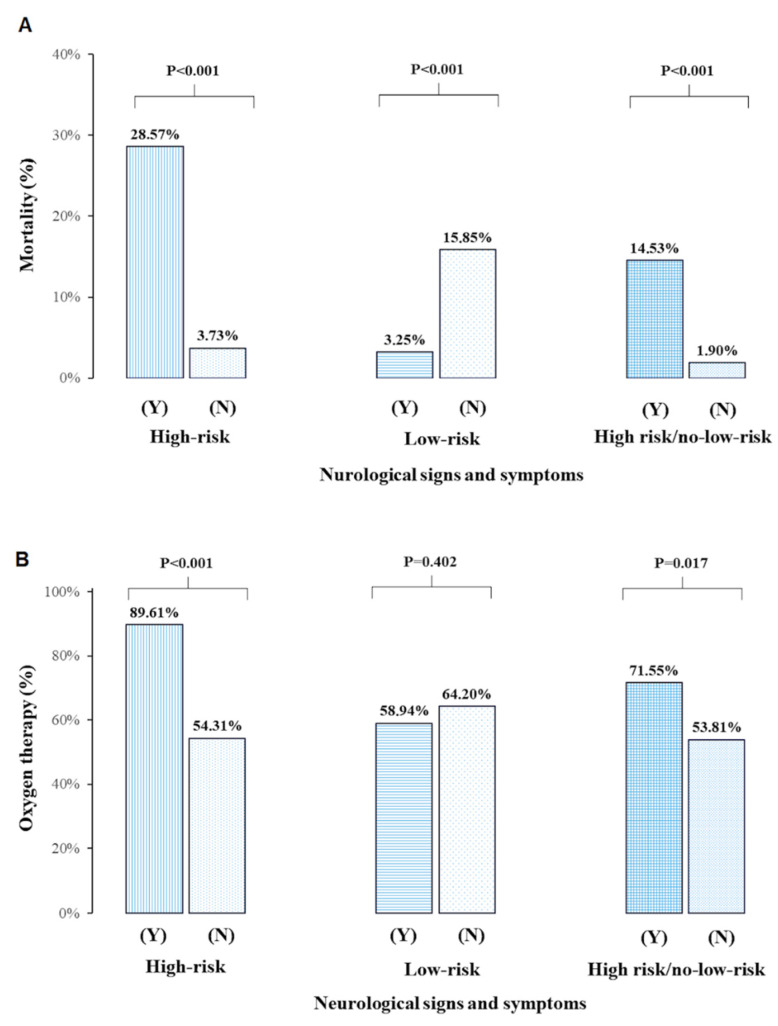
Mortality rates (panel **A**) and requirement for oxygen therapy (panel **B**) in COVID-19 patients according to the severity of neurological symptoms and signs (NSS). High-risk NSS include decreased level of consciousness, delirium, seizures, and symptoms of stroke or transient ischemic attack. Low-risk NSS comprise headache, dizziness, decreased mood, and fatigue.

**Figure 2 brainsci-12-00193-f002:**
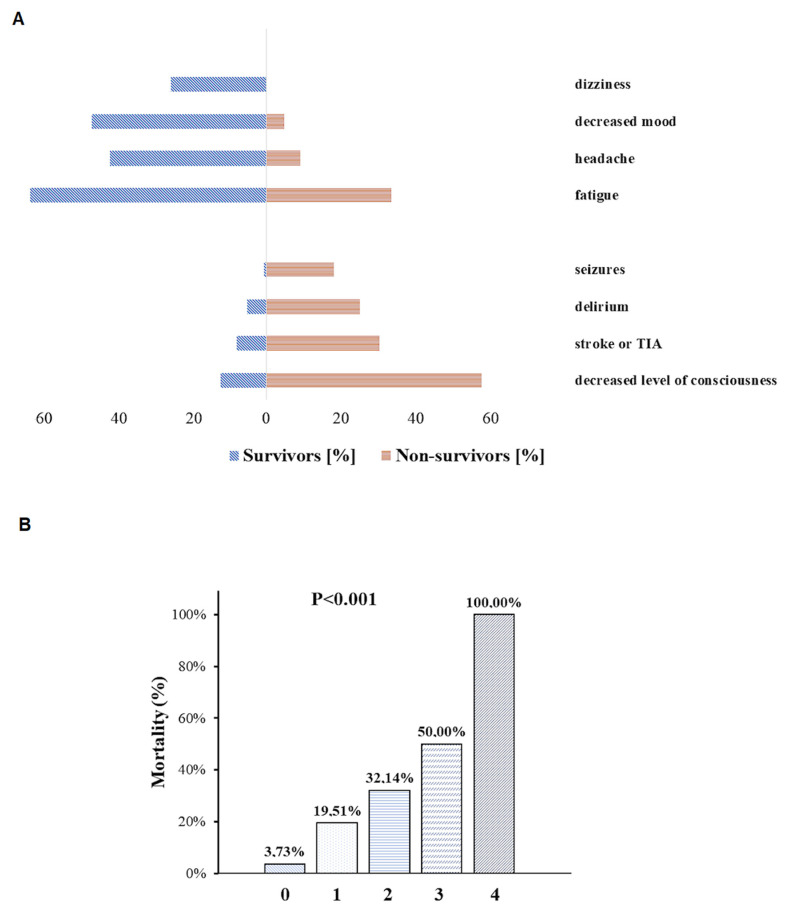
Occurrence of neurological symptoms and signs in COVID-19 survivors and non-survivors (panel **A**). Mortality rates according to the number of high-risk neurological symptoms and signs (panel **B**).

**Table 1 brainsci-12-00193-t001:** Patients’ demographic and clinical characteristics and association with death and oxygen therapy.

		Death			Oxygen Therapy		
	All Patients n = 349	No n = 316	Yes n = 33	*p*-Value	No n= 131	Yes n = 218	*p*-Value
Demographics							
Age (years)	64 (51–77)	62 (49–75)	77 (73–84)	<0.001	58 (45–69)	68.5 (55–79)	<0.001
Age >75 years, n (%)	101 (28.94)	80 (25.31)	21 (63.63)	<0.001	22 (16.79)	80 (36.70)	<0.001
Female sex, n (%)	191 (54.72)	173 (54.75)	18 (54.55)	0.982	85 (64.89)	105 (48.17)	0.002
Comorbidities and treatment							
Hypertension, n (%)	209 (59.89)	187 (59.18)	22 (66.67)	0.403	65 (49.62)	144 (66.06)	0.002
Obesity, n (%)	65 (18.62)	58 (18.35)	7 (21.21)	0.688	20 (15.27)	46 (21.10)	0.178
Smoking, n (%)	54 (15.47)	4 (14.92)	7 (21.21)	0.342	17 (12.98)	37 (17.13)	0.301
Diabetes mellitus, n (%)	90 (25.79)	76 (24.05)	14 (42.42)	0.022	26 (19.85)	64 (29.36)	0.049
Ischemic heart disease, n (%)	59 (16.91)	51 (16.14)	8 (24.24)	0.237	16 (12.21)	43 (19.72)	0.077
Prior CNS disease, n (%)	77 (22.06)	56 (17.72)	21 (63.64)	<0.001	17 (12.98)	60 (27.52)	0.002
Stroke	40 (11.46)	27 (8.54)	13 (39.39)	<0.001	10 (7.63)	30 (13.82)	0.079
Dementia	19 (5.44)	16 (5.06)	3 (9.10)	0.332	0 (0.00)	19 (8.76)	<0.001
Parkinsonian syndrome	7 (2.01)	4 (1.23)	3 (9.09)	0.029	2 (1.53)	5 (2.30)	0.714
Epilepsy	13 (3.72)	12 (3.80)	1 (3.03)	1.000	4 (3.05)	9 (4.15)	0.773
CNS tumor	8 (2.29)	4 (1.27)	4 (12.12)	0.004	1 (0.76)	7 (3.23)	0.267
Traumatic brain injury	5 (1.43)	3 (0.95)	2 (6.06)	0.072	1 (0.76)	4 (1.84)	0.654
Asthma / COPD, n (%)	20 (5.73)	20 (6.33)	0 (0.0)	0.237	4 (3.05)	17 (7.80)	0.102
Neoplasm, n (%)	40 (11.46)	31 (9.81)	9 (27.27)	0.027	6 (4.58)	33 (15.14)	0.002
Chronic kidney disease stage 3, n (%)	17 (4.87)	12 (3.80)	5 (15.15)	0.039	3 (2.29)	14 (6.42)	0.121
Immunosupressive treatment, n (%)	18 (5.16)	16 (5.06)	2 (6.06)	0.683	5 (3.82)	13 (5.96)	0.460
First COVID-19 symptoms							
Fever, n (%)	207 (59.31)	188 (59.49)	19 (57.68)	0.831	67 (51.15)	141 (64.68)	0.012
Cough, n (%)	219 (62.75)	199 (62.97)	20 (60.61)	0.788	72 (54.96)	147 (67.43)	0.019
Sore throat, n (%)	44 (12.61)	42 (13.29)	2 (6.06)%	0.405	24 (18.32)	20 (9.17)	0.013
Loss of appetite, n (%)	105 (30.09)	100 (31.65)	5 (15.15)	0.071	39 (29.77)	67 (30.73)	0.849
Dyspnea, n (%)	178 (51.00)	156 (49.37)	22 (66.67)	0.058	34 (25.95)	145 (66.50)	<0.001
Abdominal pain, n (%)	67 (19.20)	62 (19.62)	5 (15.15)	0.535	22 (16.79)	46 (21.10)	0.325
Neurological symptoms and signs							
Headache, n (%)	130 (37.24)	128 (41.69)	2 (9.09)	0.025	64 (49.23)	67 (33.67)	0.005
Dizziness, n (%)	78 (22.35)	78 (25.41)	0 (0.0)	0.032	29 (22.31)	50 (25.13)	0.558
Decreased mood, n (%)	44 (41.26)	143 (46.58)	1 (4.76)	<0.001	58 (44.62)	87 (43.94)	0.904
Memory or concetration difficulties, n (%)	57 (16.33)	50 (16.29)	7 (31.82)	0.063	20 (15.38)	37 (18.59)	0.452
Fatigue, n (%)	200 (57.31)	193 (62.87)	7 (33.33)	0.007	77 (59.23)	124 (62.63)	0.536
Visual disturbances, n (%)	26 (7.45)	24 (7.82)	2 (9.52)	0.677	13 (10.00)	13 (6.57)	0.260
Decreased level of consciousness, n (%)	57 (16.33)	38 (12.06)	19 (57.58)	<0.001	3 (2.29)	54 (24.88)	<0.001
Delirium, n (%)	24 (6.88)	16 (5.11)	8 (25.0)	<0.001	2 (1.54)	2 (10.23)	0.002
Seizures, n (%)	8 (2.30)	2 (0.63)	6 (18.18)	<0.001	0 (0.0)	8 (3.69)	0.027
Ataxia, n (%)	6 (1.72)	5 (1.61)	1 (3.57)	0.406	2 (1.54)	4 (1.91)	1.000
Involuntary movements, n (%)	16 (4.58)	12 (3.81)	4 (12.12)	0.054	4 (3.05)	12 (5.53)	0.428
Symptoms of stroke / TIA, n (%)	35 (10.03)	25 (7.91)	10 (30.30)	<0.001	6 (4.58)	29 (13.3)	0.008
Anosmia, n (%)	73 (20.92)	70 (22.80)	3 (14.29)	0.587	28 (21.54)	45 (22.73)	0.800
Ageusia, n (%)	89 (25.50)	87 (28.34)	2 (9.52)	0.075	30 (23.08)	59 (29.80)	0.181
Muscle weakness, n (%)	160 (45.85)	149 (48.22)	11 (50.00)	1.000	55 (42.31)	106 (52.74)	0.063
Myalgia, n (%)	122 (34.96)	114 (37.13)	8 (38.10)	1.000	45 (34.62)	78 (39.39)	0.381
Paresthesia, n (%)	64 (18.34)	61 (19.81)	3 (14.29)	0.776	29 (22.31)	35 (17.59)	0.290
Diarrhea, n (%)	92 (26.36)	87 (27.53)	5 (15.63)	0.145	30 (22.90)	63 (29.03)	0.210
Increased sweating, n (%)	115 (32.95)	110 (35.71)	5 (20.00)	0.13	37 (28.46)	78 (38.42)	0.062
Blood pressure <90/60 mmHg, n (%)	66 (18.91)	48 (15.19)	18 (54.55)	<0.001	13 (9.92)	53 (24.31)	<0.001
Heart rate (>100/min), n (%)	105 (30.09)	89 (28.16)	16 (48.48)	0.015	29 (22.14)	78 (34.86)	0.012
High-risk NSS, n (%)	77 (22.06)	55 (17.57)	22 (68.75)	<0.001	8 (6.15)	69 (32.24)	<0.001
Low-risk NSS, n (%)	246 (70.49)	238 (77.52)	8 (38.1)	<0.001	101 (71.69)	145 (73.60)	0.402
High-risk/absence of low-risk NSS, n (%)	117 (35.78)	100 (32.68)	17 (80.95)	<0.001	33 (25.38)	83 (42.53)	0.017
Hospital admission							
Oxygen therapy, n (%)				<0.001	-		-
Not required	131 (37.53)	131 (40.82)	0 (0.0)			150 (68.80)	
Nasal cannula	150 (42.50)	146 (46.20)	4 (12.1)			65 (29.81)	
Non-re-breather mask	65 (18.62)	36 (11.40)	29 (87.87)			3 (1.37)	
Non-invasive ventilation	3 (0.86)	3 (0.95)	0 (0.0)			(0.0)	
MEWS score, n (%)				<0.001			<0.001
0–2	330 (95.10)	306 (97.14)	24 (75.00)		131 (100.0)	198 (92.09)	
≥3	17 (4.90)	9 (2.86)	8 (25.00)		0 (0.00)	17 (7.91)	
Laboratory tests							
Troponin I (mg/dL)	6.39 (3.24–16.41)	5.80 (2.95–13.36)	21.21 (11.53–39.53)	<0.001	4.02 (1.25–8.56)	8.29 (4.39–20.27)	<0.001
D-dimer (mg/L)	0.72 (0.44–1.48)	0.70 (0.43–1.39)	1.23 (0.63–3.31)	0.005	0.53 (0.31–1.17)	0.86 (0.51–1.68)	<0.001

Values are presented as n (%), mean ± standard deviation, or median (interquartile range). COPD denotes chronic obstructive pulmonary disease; CNS, central nervous system; MEWS, Modified Early Warning Score; NSS, neurological symptom or sign; TIA, transient ischemic attack.

**Table 2 brainsci-12-00193-t002:** Multivariable regression analysis for death.

	Univariable Analysis		Multivariable Analysis					
			Model A		Model B		Model C	
	OR (95% CI)	*p*-Value	OR (95% CI)	*p*-Value	OR (95% CI)	*p*-Value	OR (95% CI)	*p*-Value
Age, decades	2.08 (1.51–2.86)	<0.001	1.70 (1.11–2.61)	0.016	1.68 (1.09–2.60)	0.020	1.58 (1.02–2.45)	0.041
Prior CNS disease	8.13 (3.78–17.47)	<0.001	5.26 (1.86–14.90)	0.002	-	-	-	-
Diabetes mellitus	2.33 (1.11–4.86)	0.025	-	-	-	-	-	-
Chronic kidney disease stage 3	4.52 (1.49–13.76)	0.008	-	-	-	-	-	-
Neoplasm	3.49 (1.47–8.08)	0.004	4.64 (1.48–14.56)	0.008	3.93 (1.14–13.59)	0.031	4.83 (1.42–16.37)	0.001
High-risk neurological symptoms or signs	10.32 (4.63–23.02)	<0.001	3.13 (1.11–8.84)	0.031	x	x	x	x
Low-risk neurological symptoms or signs	0.18 (0.07–0.49)	<0.001	x	x	0.15 (0.05–0.48)	0.001	x	x
High-risk/or absence of low-risk neurological symptoms or signs	8.76 (2.87–26.70)	<0.001	x	x	x	x	7.67 (1.94–30.20)	0.004
MEWS score (per point)	2.08 (1.51–2.88)	<0.001	2.00 (1.30–3.04)	0.001	2.25 (1.42–3.54)	<0.001	2.00 (1.29–3.10)	0.002
Troponin I (log)	1.60 (1.25–2.05)	0.002	-	-	-	-	-	-
D-dimer (log)	1.70 (1.26–2.28)	<0.001	-	-	-	-	-	-
AIC			127.21		104.75		104.73	
The Wald test, (df), *p*-value			36.02, (5), <0.001		27.10, (4), <0.001		25.73, (4), <0.001	
AUC (95% CI)			0.91 (0.87–0.95)		0.89 (0.83–0.95)		0.89 (0.83–0.95)	
The Hosmer–Lemeshow test *p*-value			0.678		0.727		0.566	

For abbreviations, see Table 1. AIC denotes Akaike information criterion; AUC, area under the receiver operating characteristic curve; CI 95% confidence intervals; (df) degree of freedom; OR Odds Ratio; x a variable not included in a model.

**Table 3 brainsci-12-00193-t003:** Multivariable regression analysis for oxygen therapy.

	Univariable Analysis		Multivariable Analysis					
			Model A		Model B		Model C	
	OR (95% CI)	*p*-Value	OR (95% CI)	*p*-Value	OR (95% CI)	*p*-Value	OR (95% CI)	*p*-Value
Age, decades	1.41 (1.2–1.62)	<0.001	1.21 (1.01–1.49)	0.042	1.29 (1.08–1.54)	0.005	1.24 (1.03–1.49)	0.020
Female sex	0.51 (0.33–0.79)	0.003	0.51 (0.29–0.91)	0.023	0.51 (0.29–0.90)	0.019	0.50 (0.28–0.87)	0.018
Prior CNS disease	2.56 (1.42–4.62)	0.002	-	-	-	-	-	-
Hypertension	1.96 (1.26–3.06)	0.003	-	-	-	-	-	-
Diabetes mellitus	1.69 (1.01–2.84)	0.048	-	-	-	-	-	-
Neoplasm	3.74 (1.52–9.18)	0.004	3.85 (1.31–11.31)	0.014	3.33 (1.15–9.64)	0.027	3.46 (1.18–10.14)	0.024
High-risk neurological symptoms or signs	7.26 (3.36–15.68)	<0.001	4.48 (1.88–10.68)	0.001	x	x	x	x
Low-risk neurological symptoms or signs	0.80 (0.48–1.4)	0.298	-	-	x	x	x	x
High-risk/no low-risk neurological symptoms or signs	2.16 (1.32–3.51)	0.002	x	x	x	x	1.86 (1.01–3.46)	0.049
MEWS score	5.01 (3.09–8,14)	<0.001	5.49 (3.05–9.74)	<0.001	5.78 (3.24–10.32)	<0.001	5.40 (3.03–9.62)	<0.001
Troponin I (log)	1.55 (1.26–1.92)	<0.001	-	-	-	-	-	-
D-dimer (log)	1.67 (1.30–2.15)	<0.001	-	-	-	-	-	-
AIC			299.33		303.45		301.21	
The Wald test, (df), *p*-value			55.35, (5), <0.001		48.14, (4), <0.001		64.03, (5), <0.001	
AUC (95% CI)			0.83 (0.79–0.87)		0.80 (0.74–0.86)		0.79 (0.75–0.83)	
The Hosmer–Lemeshow test *p*-value			0.366		0.241		0.263	

For abbreviations, see Table 1 and Table 2.

## Data Availability

The data that confirm the results of the current study are available from the corresponding author upon reasonable request.

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
