# Peer review of "Neurological Prognostic Factors in Hospitalized Patients with COVID-19"

_brainsci, 2022, doi:10.3390/brainsci12020193_

Round 1

Reviewer 1 Report

Thank you very much for allowing me to review this manuscript, which aim is to analyze whether combination of easy to identify neurological symptoms or signs (NSS) and MEWS (Modified Early Warning Score) score were associated with in-hospital mortality or oxygen requirement during the first 14 days of hospitalization in COVID-19 patients.

COMMENTS AND SUGGESTIONS

  • The abstract section must be without headings
  • The abstract section must be adapted to 250 words, according to the rules established by the journal
  • The abstract section does not sufficiently explain the methodology used.
  • The abstract section results should be summarized
  • The the introduction is too short. The current situation of the subject of study should be explained in greater depth
  • In the methodology section, the authors must specify the type of study they will carry out
  • What is the reason why the authors considered "no possibility of social isolation" as an inclusion criterion? It has no relationship with the rest of the criteria considered
  • The inclusion or exclusion of certain patients may introduce biases and reduce the validity of the results.
  • Although previously published, the authors should better explain the study variables considered and through which instruments they obtain these data.
  • The authors should better explain how the data collection procedure is carried out
  • Were the multivariate analysis models adjusted according to any variable?
  • In the results section, the authors should indicate the value of p when commenting on the statistical relationships
  • Tables should be presented as close as possible to the text in which they are referred to, not at the end of the results. Tables should be simplified for better understanding
  • The discussion section should be further enhanced and enriched
  • The conclusion of the study is very brief. It should be expanded, taking into account the results obtained

Author Response

Response to the Reviewer 1 and the Editor is attached as Word file.

Reviewer 2 Report

I read with interest the manuscript by Drabik et al., aimed at identifying new neurological prognostic factors in hospitalized patients with COVID-19. The study is quite original and well-conducted. Moreover, the sample numerosity, in line with other patients’ demographic reported in literature, and the statistical methods seem adequate for the purpose of the study. I have just few comments and suggestions, aiming to improve the global quality and readability of the manuscript:

  • Please, correct “SARS-Cov-2” into “SARS-CoV-2”
  • A better description (up to 5-6 lines) of the broad variety of COVID-19-related neurological signs/symptoms, ranging from mild to severe manifestations, should be provided in the first part of introduction, possibly referring to up-to-date reports.
  • The authors claimed there is “the need for simple and reliable prognostic factors that could be helpful in predicting the course of SARS-CoV-2 infection during hospitalization”. I completely agree with this statement. However, it should be noted that, alongside “clinical and paraclinical findings”, some relevant prognostic biomarkers, easy to obtain and endowed with prognostic value, have been already developed. Relevant to this, recent works exploring the prognostic value of monocyte distribution width (MDW) in COVID-19 hospitalized patients should be cited and briefly commented.
  • The authors mostly focused on the prognostic value of a combination of clinical neurological and non-neurological parameters, while only two biomarkers (troponin-I and D-dimer) have been considered. In the opinion of the authors, could the implementation of new biomarkers (such as MDW) within the proposed models improve their prognostic ability? A brief comment on this issue could also be appreciated.
  • The role of instrumental (EEG) or imaging (MR, CT) CNS investigations in COVID-19 patients with neurological symptoms was not discussed by the authors. Was the neurological evaluation based only on questionnaires/clinical parameters or were instrumental/imaging investigations also performed in the enrolled patients (or in a part of them)? May these diagnostic tools help to further refine the prognosis and improve the clinical management of COVID-19 patients with neurological signs/symptoms? Could they be usefully integrated in the proposed models? Although the CNS instrumental evaluation clearly goes beyond the scope of the present study, a brief discussion on this point could be interesting for the broad readership of the Journal.

Author Response

The response to the Reviewer 2 is attached as Word file.
